# Civilizational Populism in Indonesia: The Case of Front Pembela Islam (FPI)

Ihsan Yilmaz [1], Nicholas Morieson [1] and Hasnan Bachtiar [2,*]

1    Alfred Deakin Institute for Citizenship and Globalisation (ADI), Deakin University, Melbourne 3125, Australia
2    Faculty of Islamic Studies, University of Muhammadiyah Malang (UMM), Malang 65144, Indonesia
*    Correspondence: bachtiar@umm.ac.id

**Abstract:** This article examines whether a 'civilizational turn' has occurred among populist movements in Indonesia. It focuses on the civilizational elements in the populist discourse of the Front Pembela Islam (Islamic Defender Front/FPI) in Indonesia. The article traces the FPI's history and growing influence on politics and society in Indonesia in the 2010s. This article argues that the FPI has instrumentalized religious discourse, and through it divided Indonesian society into three groups: the virtuous ummah, corrupt elites, and immoral internal and external non-Muslim enemies, especially the civilizational bloc 'the West'. This instrumentalization gained the group a degree of popularity in the second decade of the post-Suharto period and strengthened its political power and ability to bargain with mainstream political parties. The article uses the FPI's actions and discourse during the Ahok affair to demonstrate the civilizational turn in Indonesian populism. The article shows how the FPI grew in power during the Ahok affair, in which a Christian Chinese politician, Basuki Tjahaja Purnama, was accused of blasphemy by Indonesian Islamists and later convicted on the same charge by an Indonesian court. The FPI was a leading part of a broad coalition of Islamist groups and individuals which called for Ahok to be charged with blasphemy; charges which were eventually laid and which led to Ahok being sentenced to two years imprisonment. The FPI, the article shows, framed Ahok as a non-Muslim Christian and therefore a 'foreign' enemy who was spreading moral corruption in Indonesia, governing 'elites' as complacent in combating immorality and positioned themselves as defenders of 'the people' or ummah. From the security perspective of the state, the FPI presented a critical threat that required containing. As a result of the growing power of the group, the FPI was banned in 2020 and Rizieq was imprisoned, while Ahok was politically rehabilitated by the Widodo government. Although the FPI's banning is considered the most effective nonpermanent solution for the state, there is evidence that the FPI's discourse has been adopted by mainstream political actors. This article, then, finds that the growth of the FPI during the second decade of the post-Suharto period, and their actions in leading the persecution of Ahok, demonstrates a civilizational turn in Indonesian Islamist populism.

**Keywords:** Front Pembela Islam (FPI); religious populism; civilizational populism; Islamism; populism in Indonesia

## 1. Introduction

In Indonesia, although discourse on the clash of civilizations has never dominated the mainstream political debate, Islamist political actors have expressed enmity toward 'the west', claimed that Western powers are working with Indonesian ruling elites to undermine Islam, and sought to unite Muslims politically. This rhetoric has been used by both violent extremist groups and nonviolent sociopolitical movements in Indonesia and throughout a variety of democratic Muslim-majority nations. For example, al-Qaeda believes that Muslims are morally good and oppressed by immoral and corrupt Muslim elites, non-Muslims, and secularist Western powers. They, therefore, wage global jihad against these representations of 'evil' (Yilmaz et al. 2021b; Zúquete 2017, p. 449). Movements of Islamist

populism, in particular in Indonesia, have adopted a similar discourse that claims 'the West' and 'elites' are working together to oppress Muslims worldwide. This argument has often taken on a religious and moral character in the Muslim world and involves the bifurcation of the world into two moral categories: good and evil.

Populism is a group of ideas that "considers society to be ultimately separated into two homogenous and antagonistic groups, 'the pure people' versus 'the corrupt elite' and/or dangerous others. It argues that politics should be an expression of the volonté générale (general will) of the people" (Mudde 2017, p. 543). According to Mudde (2017), populists often portray 'the people' as morally good, and elites as morally 'bad,' insofar as the latter has turned away from and betrayed 'the people' and their way of life. However, as a loose group of ideas rather than a 'thick' ideology, populism is most often combined with a 'thicker' ideology in order to provide it with content (de la Torre 2019, p. 7). This 'thicker' ideology to which populism adheres may be a political ideology (socialism or nationalism) or a religion. Indeed, populism may unite religion and ethnonationalism into a single political program. Populism has been combined with Islam in a wide variety of cultural and physical geographies (Zúquete 2017, p. 449; Hadiz 2018; Yilmaz and Morieson 2021, 2022a, 2022b). In each case, Muslim populist movements have used Islam to help define populism's key signifiers: 'the people', 'the corrupt elite', and 'others'. In Islamic populism 'the people' are the Muslim people of the nation, 'elites' are the insufficiently Muslim and therefore immoral ruling class, who do not respect the will of the people, and 'others' are non-Muslims, the West, and secular and liberal Muslims, who are portrayed as conspiring to oppress Muslims. Hadiz (2018, p. 567) observes that in the case of Indonesia, Turkey, and Egypt, " . . . cultural idioms associated with Islam are required . . . for the mobilisation of a distinctly ummah-based political identity in contests over power and resources in the present democratic period".

For example, Islamic populism was evident in Pakistan under the leadership of former Prime Minister Imran Khan. Khan's populist discourse divided Pakistan into three groups: corrupt and immoral elites, 'the people' (pious Sunni Muslims), and 'others' (non-Muslims, secular and liberal Muslims) (Yilmaz and Morieson 2022b; Yilmaz and Shakil 2021; Shakil and Yilmaz 2021). According to Khan, Pakistan's ruling elite was ignoring the will of 'the people' and conspiring with Western powers, especially the United States, to oppress Muslims in Pakistan. He claimed he would unite Pakistani Muslims and liberate them from Western oppression by constructing a 'New Pakistan', a Sunni Muslim heartland based on the Islamic city-state of Medina (Yilmaz and Morieson 2022b; Yilmaz and Shakil 2021).

In Turkey, populist President Recep Erdogan divides Turkish society into three categories: corrupt and immoral elites who had abandoned Islam for Western secularism, 'the people', or Sunni Muslim ethnic Turks, and dangerous 'others' (Kurds, non-Sunni Muslims, liberals, secularists) whom Erdogan alleged were working with Western powers to oppress Muslims (Yilmaz 2018; Yilmaz et al. 2021a). Following a period of Muslim democracy and pluralism, Erdogan and his Justice and Development Party embraced populism, Islamism, and a neo-Ottoman political agenda (Yilmaz 2018; Yilmaz et al. 2021a). Erdogan claims that he and his party are acting to protect the Turkish people from foreign designs on their country (Yilmaz 2021). To do this, he claims he is reviving Islamic civilization in Turkey and returning the country to its rightful place as leader of the Muslim world (Hazir 2022). Erdogan's populist discourse, like Imran Khan's, involved a civilizational classification of peoples and posited that Muslims ought to band together to protect themselves from Western attempts to dominate them.

In many respects, the Islamic populism of Khan and Erdogan reflects the wider 'civilizational turn' among populists, first noted by Brubaker (2017), Kaya and Tecmen (2019), and later described by Yilmaz and Morieson (2022b). Yilmaz and Morieson (2022a) describe 'civilizational populism' "as a group of ideas that together considers that politics should be an expression of the volonté générale (general will) of the people, and society to be ultimately separated into two homogenous and antagonistic groups, 'the pure people' versus 'the corrupt elite' who collaborate with the dangerous others belonging to other

civilizations that are hostile and present a clear and present danger to the civilization and way of life of the pure people", and find it present in the mainstream politics of India, Myanmar, and Pakistan.

In Indonesia, populism is increasingly present in the public sphere (Mietzner 2015). Yet Islamist populism has not proven as electorally successful as it has in Turkey or Pakistan. Mietzner (2015), for example, argues that incumbent Indonesian President Joko Widodo (commonly known as 'Jokowi') and Presidential candidate Prabowo Subianto may be considered populists. Jokowi, Mietzner (2015) writes, portrayed himself as a champion of the Indonesian people who would end elite corruption. Prabowo, a former General in Indonesia's military, also attacked Indonesian 'elites', but also made alliances with Islamist groups while running for President and portrayed himself as a pious Muslim. However, he stopped short of demonizing non-Muslims and 'the West' and criticized people who claim the West and the Islamic world are locked in a clash of civilizations (Mietzner 2015).

The violent Indonesian Islamist social movement Islamic Defenders Front (FPI) does not run candidates at elections, but through its street power and activism exercised significant influence over mainstream Indonesian politics throughout the 2010s, and before the group was banned and its leader, Muhammad Rizieq Shihab, imprisoned. The FPI portrays itself as a champion of the Indonesian people and an enemy of the country's allegedly immoral and secular 'elites'. The group—much like Erdogan in Turkey and Khan in Pakistan—alleges that national elites are permitting immorality to spread from the West throughout the country and that Muslims must unite as an ummah to overcome traitorous elites and Western imperialism. The FPI also claims that Indonesian minority groups, including Christians, Chinese (most of whom are Christian), Shia Muslims, and Ahmadiyya, are either spreading immorality among Indonesian Muslims or perverting Islam with incorrect teachings, and in doing so present an existential threat to the pious Sunni Muslim people. The FPI was most influential during the Ahok affair. Basuki Tjahaja Purnama (known by his Hakka name Ahok), is a prominent businessman and politician who, in 2017, was Deputy Governor of Indonesia's capital city, Jakarta. A Chinese Christian, Ahok angered many Indonesian Muslims when he criticized conservative Muslims' claims that the Qur'an demanded that no non-Muslim be permitted to rule over Muslims.

The backlash against Ahok, much of it led by Rizieq and the FPI, led to mass rallies in which Rizieq found a new and large audience receptive to his Islamist populism. At these rallies, which attracted hundreds of thousands of people, Rizieq called for Ahok to be jailed, and for a social revolution in Indonesia which would end the dominance of secular elites, curtail Western influence, and Islamize the nation. In response to a series of anti-Ahok mass rallies, President Joko Widodo began to distance himself from Ahok, and the Indonesian police and state prosecutors succumbed to public pressure and charged Ahok with blasphemy. Rizieq was called to give expert testimony at Ahok's trial, which led to Ahok being convicted and imprisoned for blasphemy. Following this decision, both Widodo and his rival for the Presidency, Prabowo, sought to ally themselves with Islamists and portray themselves as defenders of 'the people' and pious Muslims. Yet, following Widodo's victory over Prabowo in the Presidential elections of 2019, the FPI was banned by the Widodo government, and Rizieq was jailed for four years for lying about COVID-19 test results (Aljazeera 2021), while Ahok was released from prison in 2019, rehabilitated, and placed in charge of Indonesia's largest state-owned enterprise (The Jakarta Post 2019a).

Does the rise of the FPI suggest a 'civilizational turn' among Indonesian populists similar to the civilizational turn among right-wing populists in Turkey, North America, and Europe? Or does the group's banning in 2020 signal the triumph of pluralism over divisive populism in Indonesia? To answer these questions, and to understand the rise and impact of civilizational populism in Indonesia, this article examines the ideology and political activities of the Islamic Defenders Front in Indonesia.

Specifically, the article asks whether civilizational populism is present in the discourse of the FPI and especially of its leader Rizieq. We focus on Rizieq's discourse during the Ahok affair, which brought the FPI mainstream attention and greater political significance,

in order to comprehend the role of civilizational populism in Rizieq's discourse and its purposes. In particular, we examine how Rizieq constructed and delineated the boundaries of 'the people', 'elites', and 'others', and the use of religious and civilizational belonging in constructing these boundaries. Finally, the article examines the impact the FPI's activism had on mainstream Indonesian politics by investigating the actions of presidential candidates Joko Widodo and Subianto Prabowo during campaigning in 2019.

The article begins with a brief discussion of the relationship between Islamism and the State in Indonesia and describes how, while Islamist parties have consistently failed to draw support from voters, Islamism is nonetheless a growing force in Indonesian society and politics. Following this, the article examines the history and ideology of the FPI, the rhetoric of its leader, Muhammad Rizieq Shihab, describes its growth in stature and significance during the 2010s, and its impact on Indonesian politics during the Ahok affair. The article shows how the FPI uses a civilizational discourse in which Indonesian Sunni Muslims are portrayed as part of a virtuous transnational ummah, and as victims of an aggressive West working with secular Indonesian elites and religious minorities to oppress the Muslim ummah in Indonesia. Equally, the article shows that although the FPI's banning has all but destroyed the group, mainstream political actors, including President Joko Widodo, were forced to co-opt some of the FPI's discourse and create alliances with Islamists in order to prevent the group from growing in power.

## 2. Islamism in Indonesia after the Fall of Suharto

The collapse of the authoritarian Suharto regime, and the subsequent democratization process, led to a more plural Indonesia. However, it also gave rise to a number of intolerant Islamist movements. The 1999 election victory of Abdurrahman Wahid, leader of the Nahdlatul Ulama, an Islamic civil society organization, seemed to point Indonesia toward a tolerant and pluralist future. Wahid had long been an opponent, not only of the Suharto regime, but of religious bigotry within Indonesia, and the antipluralism evident in some interpretations of Islamic doctrine. In 1985, as leader of NU, Wahid told his followers that an "Islamic revival" was taking place in Indonesia, and that "the richness of [Islam's] heritage from its deep perception of a true place for humanity in life, to its great tolerance, make a strong base for the Muslims to sail through the revival process" (The Jakarta Post 2019b). Wahid "conceived of Islam as a cosmopolitan civilization, which contains many cultures and different ideas which come from different sources" (Barton et al. 2021a; Mujiburrahman 1999, p. 346; Wahid 2007). Thus, he believed that democracy and mild secularism or nonsectarianism could flourish in Islam, even if they were originally somewhat 'foreign' concepts since they were harmonious with the core teachings of Islam, which support pluralism and government by the consent of the people (Mujiburrahman 1999, p. 346; Wahid 2007). Following the events of 11 September 2001, Wahid sought to downplay a 'clash of civilizations' between the West and Islam and argued that "Islamic militantism is caused by a misunderstanding of religion and a kind of inferiority complex towards Western civilisation" (Sydney Morning Herald 2002). Wahid further argued that the 'clash of civilizations' thesis is rooted in a double standard being applied to Muslims, in which any violence committed by Muslims is portrayed as Islamic in nature, and while he admitted differences between Muslims and the secular West, he suggested that "differences do not mean enmity and clashes" (Sydney Morning Herald 2002; Wahid 2003, p. 155).

While his presidency was short lived, the influence of Wahid's conception of Islamic civilization as tolerant and plural, and the overall multiethnic and multireligious nature of Indonesian society has encouraged most Indonesian Islamic organizations—especially the two largest organizations Muhammadiyah and Nahdlatul Ulama—to encourage their followers to practice religious tolerance (Barton et al. 2021a). Part of this encouragement of a plural, cosmopolitan Islam comes in the form of support for Pancasila, Indonesia's governing principles enshrined in the nation's constitution. Pancasila, meaning five principles, is not a secular doctrine but instead insists that Indonesians believe in only one God (Barton et al. 2021a; Latif 2011). Yet Pancasila does not demand that Indonesians believe in Islam,

only that they do not worship many gods or become atheists. Of course, in an environment in which around 86% of the population is Muslim, it is not surprising that Islam has a privileged place in Indonesia (Barton et al. 2021a). Yet Hinduism, despite its many gods, is acknowledged as one of the six religions (alongside Buddhism, Confucianism, Islam, and Protestant and Catholic Christianity) recognized by Indonesian law, and is interpreted as a monotheistic faith, and Orthodox Christian Churches exist and are tolerated. Therefore, it is possible to say that a significant degree of religious freedom exists in Indonesia, although it must also be admitted that religious and racial intolerance exists (Setara Institute 2022; Wahid Foundation 2019), especially intolerance toward the Ahmadiyya community (Yosarie et al. 2021, pp. 6–58).

A complex and contradictory picture, then, emerges when we examine Islam's role in contemporary Indonesia. On the one hand, Sunni Islam is the dominant faith, and the people of other religious denominations face difficulties, and at times violence, when they attempt to construct places of worship or recruit new followers. On the other hand, Muhammadiyah and Nahdlatul Ulama, two civil religious organizations so large that perhaps a third of all Indonesians are affiliated with one or the other, have since the end of the Suharto regime propagated a message that Islamic civilization tolerates religious differences and that violence against minority groups is antithetical to the teachings of Islam (Barton 1995; Fealy and Barton 1996). Within this basic framework, members of Muhammadiyah and Nahdlatul Ulama possess different notions of how Islam should be practiced and the degree to which Arab customs ought to be adopted or disregarded. Equally, the notion that there is a clash of civilizations between Islam and the West is downplayed even by politicians linked to Islamism such as Anies Baswedan (The Bali Times 2008).

### 3. Islamism and Populism in Indonesia

In the reformation era of Indonesia, a number of long-suppressed Islamic and Islamist groups gained political significance. The first postauthoritarian President, B.J. Habibie, sought to end the violent chaos erupting throughout the country by permitting religious organizations and their related militias to police the streets and counter nationalist militias supported by pro-Suharto military elites (Hadiz 2016, p. 154). This helped Islamic political and civil society organizations increase their public profiles, and helped to explain why 20 of the 48 parties that ran in the 1999 general election were Islamic parties (Al-Chaidar 1999; Adiwilaga et al. 2019). These Islamic parties, and the Islamic civil society organizations Muhammadiyah and NU, held widely different views on how Islamic civilization ought to be organized in Indonesia. While Muhammadiyah and the NU (whose leader, Abdurrahman Wahid served as Indonesian President from 1999–2001) argued persuasively that Islamic civilization permitted religious freedom and therefore supported Pancasila, a position they maintain to this day. Indeed, it remains largely in the interests of these two organizations to support religious pluralism in a society in which no single understanding of Islam and its correct practices exists.

Other Islamic groups, however, possess more ambivalent positions on the question of whether Islamic civilization permits religious freedom. For example, one of the Islamic bodies that most benefitted from the transition away from the Suharto regime was the Council of Indonesian Ulama (MUI—Majelis Ulama Indonesia). The MUI, which is essentially the peak clerical Islamic body in Indonesia, was created by Suharto in 1975 in order to control Islam. After the fall of his regime, the MUI increased in significance under the Presidency of Susilo Bambang Yudhoyono (2004–2014). Indeed, after 2004, the MUI, far from a tool of the state, became an overtly political organization in its own right. Moreover, the MUI is increasingly dominated by conservatives who do not believe that Islamic civilization permits religious freedom for non-Muslims (or indeed nonorthodox Sunni Muslims), or that non-Muslims ought to be permitted to hold positions of power over Muslims (Barton 2021). For example, the MUI has increasingly given opinions antithetical to Indonesia's nonsectarian constitution, "forbidding inter-faith joint prayers" and "condemning inter-

faith marriages" (Barton 2021). Equally, the MUI has described Indonesia's Ahmadiyya Muslims as apostates (murtad) and called upon the government to take action to prevent their communities from practicing their form of Islam (Barton 2021; Zulian 2018).

While these fatwas are not legally binding, because they come from the peak clerical body they remain of great significance in Indonesian society. In 2005, the increasingly conservative MUI gave a fatwa declaring secularism, pluralism, and liberalism incompatible with Islam (Van Bruinessen 2006; Zulian 2018). According to Van Bruinessen (2006), the MUI fatwa was "ostensibly a frontal attack on the small group of self-defined 'liberal' Muslims of Jaringan Islam Liberal (JIL, Liberal Islam Network) centered around Ulil Abshar Abdalla but was intended to delegitimise a much broader category of Muslim intellectuals and NGO activists, including some of the most respected Muslim personalities of the previous decades". The political significance of the MUI has grown over time, and the body appears to exert power over the state and judiciary. For example, the MUI played an important role in preventing President (and NU leader) Wahid's plan to revoke Indonesia's blasphemy laws. Wahid's plan failed, and since 2009 there have been at least 200 criminal convictions for blasphemy in Indonesia (Nardi 2019). The increasing number of blasphemy convictions suggests that the Indonesian judiciary is increasingly bowing to pressure from Islamists who claim that the pious "people" of the nation need "protection" from the non-Muslim blasphemers who are attempting to injure the ummah (Lindsey 2012).

The power of MUI fatwas and Islamist activism shows the growing influence of Islamism within Indonesian society (Schafer 2019). At the same time, Islamist political parties have so far failed to increase their representation in the Indonesian parliament. However, this may be because mainstream parties have co-opted, to varying degrees, Islamist rhetoric. The anti-Ahok protests of 2016 were a demonstration of the growing power of, and the inability of mainstream Indonesian political parties to push back against, Islamism in Indonesia. The protests were led and actively encouraged by a number of Islamist groups including the FPI, but hundreds of thousands of ordinary Indonesian Muslims joined in to call for the prosecution of Ahok over alleged blasphemy. The respective leaders of the NU and Muhammadiyah were disapproving of the protests, however, they did not make strong calls for their members to support Ahok or stand aside from the protests (Barton 2021).

Populism, sometimes Islamist populism, has become an increasingly powerful force over the past two decades in Indonesia. According to Indonesian scholar Marcus Mietzner (2015), the 2014 election, won by Joko Widodo, who defeated Prabowo Subianto by winning more than 53% of all votes, pitted two forms of populism against one another. Prabowo Subianto offered a "textbook" populism in which he railed against elites and foreigners, whom he claimed were working together to steal Indonesian wealth. Widodo, however, Mietzner (2015) argues, represented a technocratic populism that did not demonize any particular group (including foreigners) but instead promised reform of every area of policy in order to improve society. However, Widodo (or 'Jokowi') portrayed himself as a humble man of the people who could support the lower middle classes and improve their lives. Widodo succeeded, in Mietzner's (2015) view, since there was no great crisis in Indonesia in 2014 for a confrontational populism to exploit. Technocratic populism was more appealing to voters who wanted more democracy, but who were not concerned about existential crises engulfing the nation. However, he notes, 47% of voters wanted the authoritarian and radical Prabowo (Mietzner 2015).

Violent populist Islamist groups have also emerged in postdemocratic Indonesia. Indeed, there are Islamic groups in Indonesia that do not share with Muhammadiyah and Nahdlatul Ulama the notion that Islamic civilization is plural and permits a degree of religious freedom. Nor are they concerned about the consequences of inflaming religious and ethnic tensions or asserting that the ummah is threatened by non-Muslim forces and must defend itself. As Barton et al. (2021b) observed, "since the denouement of the Suharto regime in May 1998, religion and populism have become dominant political and social forces in Indonesia". In particular, Islamist populism has become a powerful social

force, particularly in areas where the influence of Muhammadiyah and the Nahdlatul Ulama is absent or waning. Perhaps the most politically, though not electorally, successful populist Islamist movement in Indonesia is the Islamic Defenders Front (FPI). One of the first violent Islamist groups to organize following the fall of the Suharto regime was the Front Pembela Islam (the Islamic Defenders Front/FPI). Formed by Muhammad Rizieq Shihab in 1998, the FPI established itself as a proviolence, far-right movement dedicated to Islamizing Indonesia. Indeed, the FPI was frequently involved in a kind of vigilantism they portrayed as acts necessary to protect the ummah and Islam from "vice" (Barton 2021; Amal 2020; Fossati and Mietzner 2019; Mietzner 2018). These violent acts, which ranged from attacking nightclub patrons, venues serving alcohol, and brothels, to assaulting Ahmadiyya Muslims and destroying their places of worship, have brought the FPI a reputation as a dangerous and aggressive group (ABC News 2020). Yet the FPI cannot be considered a fringe movement. Through violent acts, protests, physical intimidation, and the populist rhetoric of leader Muhammad Rizieq Shihab, the FPI gained a large following among poor and conservative Muslims and exerted considerable influence at times over mainstream political parties. Indeed, the group's influence grew so great that in 2020 the Widodo-led government banned the group and jailed Rizieq.

## 4. Front Pembela Islam (FPI) History and Ideology

The FPI is a far-right, populist social and political organization dedicated to the Islamization of Indonesia. While nowhere near as popular as the leading Islamic organizations in Indonesia, Nahdlatul Ulama and Muhammadiyah, who combined possess at least 70 million members (Al-Ansi et al. 2019), the FPI claims to have seven million members (Fealy and White 2021) and enjoyed an outsized influence over Indonesian politics in the 2010s due to its ability to instrumentalize religion and portray Indonesia's governing and business elites as corrupt and un-Islamic (Barton 2021). The FPI's core message is that the ummah within Indonesia is threatened by secular, liberal, and non-Muslim forces which conspire to corrupt the morals of the ummah (Barton 2021; Fossati and Mietzner 2019; Aspinall and Mietzner 2019; Rosadi 2008; Syihab 2008). Indeed, the FPI promises a "moral" solution to Indonesia's many social problems, or the nation's "waywardness", as the FPI put it (Barton 2021; Fossati and Mietzner 2019; Aspinall and Mietzner 2019; Rosadi 2008; Syihab 2008). According to Mietzner (2020, p. 425), radical Islamist groups such as the FPI assert that "pious" Muslims "are victimised, in Indonesia and elsewhere, by non-Muslim or otherwise sinful forces, mostly in the West but also, increasingly, China. For the Indonesian context, this means that devout Muslims are kept away from power through an inter-connected conspiracy by non-Muslim countries and Indonesian elites". In this way, the FPI suggests that Muslims, as a religious-civilizational bloc, face constant attacks from non-Muslim enemies, both inside Indonesia and across the globe.

The FPI was founded in 1998 by Muhammad Rizieq Shihab, a cleric inspired by Salafism, an ideology which had evolved since the 1980s, although it had been constrained by Suharto's political interest, who serves as the group's Imam Besar (the Grand Imam). The group began its life as a vehicle for Rizieq's ultra-conservative Islamism and became involved in street violence and attacks on Muslim sects Rizieq perceived as blasphemous or insufficiently Islamic (Barton 2021). The FPI's founding purpose was "amar ma'ruf nahi munkar, or 'commanding right and forbidding wrong'," (Bamualim 2011, p. 270) and "has assembled a jemaah (community of followers) to manage religious activities and . . . mobilized laskars (soldiers) to enforce amar ma'ruf nahi munkar" (Bamualim 2011, p. 267). Essentially, the FPI believes that because the Indonesian state consistently fails in its duty to 'command right and forbid wrong', the FPI and its supporters are compelled by their religion to take the required action (Ugur and Ince 2015, p. 35; Republika 2020). Yet the FPI is not merely a violent vigilante group. It also provides welfare to Indonesia's urban poor in the form of schools, food, and employment (Jahroni 2004; Hookway 2017). Mixing welfarism with Islamic ideas of social justice, the FPI claimed its violent attacks on venues serving alcohol or permitting gambling, and on Ahmadiyya and Shia places of

worship, were justified acts required to defend Islam and the ummah (Gani 2011; Barton 2021). Moreover, like other populist groups, the FPI frames itself as 'saving the people' from an existential threat, in this case emanating primarily from the secular and Christian West and secular and non-Muslim elites within Indonesia. This is a powerful narrative that has helped the group amass a significant degree of political influence (Mietzner 2020; Peterson 2020; Waty 2020).

Ugur and Ince (2015, p. 42) observe "three major causes" behind the rise of the FPI in Indonesia: "the perception that Islamic faith is threatened by global and local forces and the faith should be protected, the demand that Islamic Sharia's 'universal' laws should be implemented and enforced by the state, and the claim that they support state's law enforcement officers in the fight against immorality, misdeeds and big sins". The FPI capitalizes on the perception that Islam and the ummah require defending, not merely in Indonesia but worldwide, by claiming that the group will protect Islam and Muslims from the internal and foreign forces that wish to do them harm (Barton 2021; Mietzner 2018; Hadiz 2016, p. 112; Wilson 2015). The FPI does not focus on winning seats in parliament. Instead, it attempts to influence sitting members to adopt more extreme positions in line with their particular brand of Islamism, often through acts of intimidation. Because they consider themselves as acting in God's name, the FPI is unconcerned with breaking non-Islamic Indonesian law, and with ignoring government authority, or declaring a sitting government illegitimate (Barton 2021). Moreover, the FPI's conception of Islam is deeply influenced by Salafism and therefore conceives of Islamic civilization as a narrow space that permits few non-Islamic influences and forbids non-Muslims from attaining high-level positions in the government.

Indeed, the FPI divides society in an Islamist-populist manner between the ummah and non-ummah. The majority population of Indonesia, observant Sunni Muslims, are the ummah, while secularists, government 'elites', non-Muslims, Ahmadiyya, and Shia Muslims, are non-ummah. Thus, the FPI frames society via a religion-based system of categorization, which denies the non-ummah a role in the public sphere, forbids the public expression of their beliefs and makes the will and interests of the ummah sacrosanct.

## 5. The FPI's Construction and Populist Instrumentalization of the Ahok 'Crisis'

The influence of the FPI grew during the 2010s when the group became a powerful presence in the Defending Islam Movement (Aksi Bela Islam/ABI) and National Movement to Safeguard the Indonesian Ulema Councils Fatwa (Gerakan Nasional Pengawal Fatwa MUI/GNPF MUI). ABI, which was a coalition of far-right groups dedicated to Islamism and which included both the FPI and Hizbut Tahrir Indonesia (HTI), was formed to protest Ahok's 'blasphemous' remarks on the misuse of the Qur'an by Islamist activists and politicians (Maulia 2020; Nuryanti 2021; Adiwilaga et al. 2019; Fossati and Mietzner 2019; Hadiz 2018; Mietzner 2018). Basuki Tjahaja Purnama (most commonly known by his Hakka nickname 'Ahok'), was accused in 2016 and later convicted of blaspheming against Islam. When a video of Ahok criticizing the manner in which certain Qur'anic verses were being misused to give the impression that Muslims cannot live under non-Muslim rule went viral online (Viva 2016), the FPI seized on the video, distorted Ahok's message, and claimed he was blaspheming against Islam (CNN Indonesia 2016; Nuryanti 2021). Indeed, many who viewed the video believed Ahok was insulting their faith, an interpretation encouraged by the FPI (Nuryanti 2021; Mietzner 2018). Despite Ahok attempting to clarify his statements, the public mood turned on the governor. Large protests erupted, and perhaps due to the size of the crowds and the growing conservatism of the body, the MUI declared Ahok's remarks blasphemous and offensive (Official NET News 2016), and essentially a crime under Indonesian law (Nuryanti 2021; Mietzner 2018). Over the next few months in 2016, more rallies against Ahok were held, culminating in a mass rally in November at which Rizieq addressed the crowd.

Hundreds of thousands of Muslims, including members of Muhammadiyah and Nahdlatul Ulama, attended these rallies during which speakers called for Ahok to be

punished by the state for his alleged crime (The Guardian 2016; Harahap and Sardini 2019). The pressure applied by these protests almost certainly led to the police charging Ahok with blasphemy and placing the governor on trial for his alleged crime (Nuryanti 2021). Ahok was convicted, in 2017 of blasphemy by an Indonesian court (CNN Indonesia 2017). According to Peterson (2020, p. 110) "throughout Ahok's case, key MUI figures, including Ma'ruf Amin, were able to use the MUI moniker with impunity. They successfully influenced law enforcement officials to arrest, indict, and convict Ahok of blasphemy, and they did so notwithstanding the fact that MUI is a non-elected body—a QUANGO—that issues legal opinions". Peterson's comments show how powerful the MUI has become in Indonesian society, and how it reaches deep into the judicial system. Equally, given the FPI's influence over MUI's decision to condemn Ahok as a blasphemer, it is difficult to deny the driving role the group played in Ahok's conviction and imprisonment. The FPI might have been just one member of a broad anti-Ahok coalition, known as the National Movement to Guard the MUI Fatwa (GNPF-MUI), which included members of Muhammadiyah and NU, Hizb-ut Tahrir, and smaller Islamist groups (Fealy 2016), however, they were the face of the movement, and coined its motto of "Defending Islam" (Fossati and Mietzner 2019, p. 774). Moreover, Ritonga et al. (2020) provide evidence that certain Indonesian newspapers portrayed the anti-Ahok rallies as part of a clash of civilizations occurring between Islamic civilization (represented in the rallies by Muslim Indonesians) and the Western and Sinic civilizations, respectively (represented by Ahok). They describe how Ahok was presented in the Indonesian media simultaneously as a Chinese and a Western threat to Islam, on the basis that he is Chinese yet also a Christian, and therefore religiously aligned to the West while being ethnically linked to China (Ritonga et al. 2020).

## 6. The Influence of the FPI's Civilizational Populism on Mainstream Indonesian Politics

Ahok's downfall had a profound effect on the Indonesian Presidential elections in 2019. Fearful of being associated with Ahok and eager to capitalize on the anti-Ahok feeling, both the incumbent President Widodo and the leading opposition figure Prabowo Subianto aligned themselves with well-known Muslim leaders. Moreover, Subianto attempted to ingratiate himself with the FPI and portrayed himself as a keen supporter of the anti-Ahok movement. For example, during his campaign, Subianto claimed that Muslim terrorists in Indonesia were, in fact, victims of poverty and foreign non-Muslim forces who were oppressing Muslims (Metro TV News 2019; Kennedy 2019). Joko Widodo, keen to shed his association with Ahok, chose NU cleric Ma'ruf Amin to be his running mate (Arifianto 2019). Widodo, however, appears to have grown increasingly concerned by the influence of the FPI. When Rizieq fled Indonesia in 2017, after being charged with pornography charges and "insulting the official state ideology, Pancasila" (Karmini 2020), it was a former Vice President, Jusuf Kalla, who helped Rizieq return from exile in Saudi Arabia in 2020 (Wirajuda 2020). However, the increasingly powerful influence of the FPI ultimately led to the group's downfall. While the Widodo government perhaps did not have the required political capital to ban the group over its campaign of violence and intimidation against non-Muslims, it found a pretext in Rizieq's decision to hold an illegal mass rally during the COVID-19 pandemic.

Rizieq's return to Indonesia coincided with the COVID-19 pandemic. When he began claiming his group would begin a "moral revolution" and staging mass rallies as part of the 212 Movement, the government responded by demanding that Rizieq obey the new laws which compelled him and his followers to be tested for COVID-19 (Majlis Alhanis 2020). Rizieq refused, and when FPI members were later found to be spreading the virus, the government decided to ban the group outright, arresting Rizieq after a supposed "encounter" between the FPI militia and Indonesian police (Kelemen 2021). In 2021, Rizieq was sentenced to four years in prison for "announcing false information and purposefully causing confusion for the public" (Reuters 2021; Pikiran Rakyat 2021). Following the group's banning, some members of the FPI regrouped under the name

"Islamic Brotherhood Front (Front Persaudaraan Islam)" and claim to be a 'moderate' organization aligned with the NU and which is willing to give conditional support to Pancasila (Tsauro and Taufiq 2022). However, the leader of the Islamic Brotherhood Front is Muhammad bin Husein Alatas, the son-in-law of Habib Rizieq Shihab, which may indicate that Rizieq will remain an influence over the group well into the future. On the other hand, the group is apparently struggling to recruit members, suggesting that its new 'moderate' face does not appeal to many former FPI supporters (Tsauro and Taufiq 2022).

## 7. Understanding the Role of Civilizational Populism in FPI Discourse

Civilizational populism plays an intriguing role in FPI discourse. The group combines populism and Islamism in a manner that requires the division of Indonesian society between two key antagonistic groups: ummah and non-ummah. This concept of 'ummah' is very important to the FPI and lies at the core of their civilizationalism. The ummah is the international brotherhood of Muslims. The Muslim ummah is, by its nature, a transnational body, spread over the entire world. By constructing the key division in Indonesian society as a battle between the ummah and non-ummah, the FPI portrays politics as a battle between the cosmic forces of good (associated with the ummah) and evil (non-ummah). By doing this, they can portray themselves and their supporters as godly and good, while government and business elites, non-Sunni Muslims, and non-Muslims are evildoers who threaten the ummah. This does not mean that the FPI is fighting to establish, in the manner of al-Qaeda and Hizb al-Tahrir, a global caliphate. While some of its members have fought for international Islamist terror groups associated with global jihad, this does not represent the mainstream tendency of the FPI as an organization. It is a populist-nationalist group concerned mostly with turning the semisecular Indonesian democracy into an Islamic political system (Kompas TV 2020; MEI@75 2021; Idris 2018, p. 9). As populists, the FPI endorses democracy. Furthermore, the group disavows plans to revive a global Islamic state or Caliphate (Fealy 2016). However, the group was found by an Indonesian court to have illegally "established ties with terrorist organization the Islamic State of Iraq and Syria (ISIS) and incited people to pledge allegiance to then the group's leader Abu Bakar Al Baghdadi" (Jakarta Globe 2022).

Their civilizationalism is therefore distinct from the civilizationalism of the global jihad movement and its attempts to create a global Islamic Caliphate and takes the form of a religious populism that divides Indonesian society between ummah and non-ummah. In this way, it is in certain respects similar to the civilizational populism of a number of European right-wing populist parties including the French National Front and the Alternative for Germany, which define their societies as inherently Christian or Judeo-Christian, and use this narrow definition to defend the exclusion of Muslims from European society (Kaya and Tecmen 2019; Kaya 2021; Morieson 2021; Marzouki et al. 2016). These parties construct a crisis-driven victimhood narrative in which Judeo-Christian civilization is faced with an existential threat from Islam and therefore argue that Muslim immigration must cease in order to protect Judeo-Christian civilization within 'our' nation (Kaya and Tecmen 2019; Kaya 2021; Morieson 2021; Brubaker 2017; Marzouki et al. 2016).

The FPI constructs a similar yet inverted argument. They claim that the ummah is everywhere under attack by 'the West' and that the Indonesian ummah is faced with a threat from 'immoral' Western culture. The anti-Westernism of the FPI is evident in the group's opposition to secularism, pluralism, and freedom of religion (Saipul 2011). The group claims that these ideas are foreign to Islam and have resulted in "many heresies and immoralities" (Ugur and Ince 2015, p. 47). Globalization is also perceived as a threat to both Indonesia and the ummah, insofar as it spreads Western liberal values the group holds to be inimical to Islam across the world, including in Indonesia Ugur and Ince (2015, pp. 42–43). Equally, globalization is perceived by FPI supporters as a kind of Western colonialism, and a "form of political, cultural, and social pressure to implement a new system/value that is clearly not suitable to be introduced to Indonesian society as the majority of our population is moslem" (Ugur and Ince 2015, p. 43). Thus, the FPI calls for the implementation of Sharia

law for Muslims are an effort to prevent the spread of ideas and immoralities associated with Western civilization.

It is possible to question whether it might be better to describe the FPI as religious nationalist populists rather than civilizational populists. However, once rhetoric identifying 'the people' as 'ummah' and calling for this ummah to unite against 'the West' is brought into populist discourse in a Muslim majority nation, it lends religious nationalist movements, such as the FPI, an inherently civilizational quality. There is no simple demarcation point between ethnoreligious/religious nationalist populism and civilizational populism. Indeed, civilizational populism is frequently a form of ethnoreligious/religious nationalist populism, and populism itself in the Islamic world is frequently aligned with ethnoreligious nationalism. Yet, due to the combination of the emphasis the FPI places on uniting 'the people' as an ummah, and its explicitly anti-Western political agenda, which characterizes the West as a civilizational bloc that is attempting to pervert the morals of the ummah, it is possible to describe the FPI as not merely a religious nationalist populist group, but as a civilizational populist group.

The FPI does not seek to create either an Islamist dictatorship within Indonesia or to fight for a global Caliphate. Instead, their goals are more modest and fall within the bounds of democracy. They do not argue that the Qur'an forbids democracy, only that Muslim-majority societies require (male) Muslim leaders (Ugur and Ince 2015, p. 52). To understand the civilizationalism in the FPI's religious populism, it is useful to examine the rhetoric of its leader and figurehead, Muhammad Rizieq Shihab, which combines populism and Islamism. Rizieq's authority stems in part from his qualifications in Islamic Law, which he studied at the Islamic University of Imam Muhammad ibn Saud, but also from his self-portrayal as a simple and pious Muslim who cares about the interests of the poorest members of the ummah (Bamualim 2011, p. 269). At the same time, Rizieq's appeal lies in his explanation of politics as a battle between good and evil. Rizieq argues that Muslims must forbid evil and command good (*al-amr bi al-ma'ruf wa al-nahy 'an al-munkar*), and this directive lies at the heart of his political discourse (Widiyanto 2017, p. 93). The phrase itself is found in the Qur'an in several places, including in verses 3:104 and 3:110, and commands Muslims to take action to prevent evil or vice from taking place. Rizieq believes that it is the religious duty of Muslims to fight evil, which he appears to conceive of as a "social pathology" affecting Indonesian society, and which the FPI will cure through its activism (Widiyanto 2017, p. 105). This is not to say that Rizieq calls for violence against all wrongdoers. Rather, he finds in the Qur'an and the Hadiths examples of violence correctly used to prevent 'vice', but also examples of nonviolent ways of combating evil (Widiyanto 2017, p. 103).

Rizieq gained followers through a populist discourse that emphasized the victimhood of the ummah (Sunni Muslims), and their oppression at the hands of the Indonesian government, various non-Sunni Muslim minorities, and beyond this by Western (and by extension Christian) civilization (Ugur and Ince 2015). In order to evoke feelings of anger and fear within his supporters, Rizieq has attempted to ensure that FPI "followers [were] kept constantly anxious about threats to their faith and way of life, and thus incentivized to hate "the Other" and at times manifest that hatred and insecurity in acts of intimidation, symbolic violence and hate speech toward out-group members" (Barton 2021). FPI followers are taught that evil is flourishing in post-Suharto Indonesia, where "Western decadence, secularism, liberalism and immorality" is permitted under the law (Bamualim 2011, p. 272). This evil takes the form, according to Rizieq and the FPI, of the "uncontrolled spread of businesses 'peddling in vice,' such as discos, bars, entertainment centres and other fronts for pornography, prostitution and illicit drugs" (Bamualim 2011, p. 272). The spread of Western decadence, according to Rizieq, has led to "a general breakdown in the moral fabric of society" (Bamualim 2011, p. 272). Yet this has not occurred, he says, by accident, but is rather as the product of non-Muslim groups "with a vested interest in the success of the businesses to bring about the gradual decline and moral decay of Islamic society" (Bamualim 2011, p. 272). In using this rhetoric, Rizieq attempts to make his supporters

fearful of the flourishing of Western influence in Indonesia, which he portrays as a crisis affecting the nation, and angry at the Indonesian government for not protecting Muslims from this 'evil' by insisting that all Muslims abide by sharia law.

In response to the spread of Western influence and evil, Rizieq calls on his followers to first seek out the evil, and then attempt to combat it by nonviolent means. The FPI, according to Rizieq, first attempts to "enlighten" people about the "noble message of Islam" and in doing so encourages them to change their ways (Widiyanto 2017, p. 106). If a community wishes for the FPI to take violent action to root out the evil within their community, then, Rizieq says, his group "is obliged to assist the local community" in destroying foreign Western practices and their associated acts of evil (Widiyanto 2017, p. 107). To people who criticize the FPI's vigilante attacks on bars, clubs, and places of worship, Rizieq argues that "evil itself is a kind of violence that does harm to people's morality, which is more valuable than property" (Widiyanto 2017, p. 103). At the same time, Rizieq frames the Indonesian government as out-of-touch elites who have failed to improve the lives of Indonesia's poorest people and demands that his followers reject the 'present situation' and instead take action to help the poor where the government will not (Bamualim 2011, p. 272). This framing allows Rizieq to portray the government as consistently failing the ummah, allowing foreign evils to prosper while the ordinary people remain impoverished, and in doing so helps him generate feelings of anger toward Indonesia's governing elites. Moreover, Rizieq portrays the FPI's illegal violent acts as lawful under sharia and claims that "true Mu'min [pious Muslim] must reject secularism, pluralism, liberalism, LGBT, apostasy, heresy, shamanism, corruption, khamr, drugs, gambling, prostitution, adultery, pornography, pornoaction, injustice, tyranny, immorality, evilness, and leadership of a kafir over Muslims, even when the constitution permits it because Qur'an and sunnah forbid it" (Sejati 2014). Thus, by invoking religion to reinforce his group's anti-government agenda and vigilante violence, Rizieq attempts to generate among his supporters the necessary antielite anger and religious rage required to create demand for the FPI's Islamist populism. Moreover, the group's vigilantism is an extension of its populist division of Indonesian society. For example, when the FPI militia attacks the worship places or businesses of minorities, their leaders categorize the attacks as necessary acts that 'defend Islam' and the ummah in a country failed by its government which has, in the post-Suharto Reformasi era, proven unwilling to command good and forbid evil.

Calls for jihad or a struggle against infidels are an important part of Rizieq's discourse. The FPI leader, for example, has declared that "it is obligatory for Muslim to unite themselves, to unite all potential" in order to make certain that Indonesia "cannot be taken by infidels!" (Aswaja 2016; Pecinta Ulama 2020). This language is designed to make Indonesian Muslims fearful and angry that non-Muslims are attempting to 'take over' Indonesia by spreading the non-Islamic values of Western civilization throughout the country, and by doing so corrupting the morals of Indonesian Muslims. Indeed, according to Rizieq, 'infidels' are lying to Muslims and tempting them to engage in Western immoral practices. He tells his supporters that "the enemy uses the weapons of lies" and therefore Muslims must "use the weapon of honesty and truth" (Aswaja 2016; Aqielabdurrani 2021). In language designed to encourage, if not compel, FPI members to fight Western 'decadence' and 'vice' in Indonesia, Rizieq rhetorically asks his supporters "if the kuffar and hypocrites are so strong in attacking Islam, why are we afraid to defend Islam?" (Aswaja 2016; Aqielabdurrani 2021). Indeed, Rizieq extols the virtue of defending Islam by fighting infidels, even if it requires one's own death. Perhaps anticipating that his supporters will be punished for using violence against minorities, or criticized for dismissing Pancasila as insufficiently Islamic, Rizieq instructs his supporters to "strengthen your heart not to break easily, so that you say istiqomah fisabilillah" (steadfast in the cause of Allah) (Sofyan 2018; Aqielabdurrani 2021). Victory, he admits, is hardly assured, and the faithful have an "obligation to fight" and to "struggle" (Sofyan 2018; Aqielabdurrani 2021). Thus, the fight to stop infidels from corrupting the morals of Indonesian Muslims is above all a spiritual and intellectual war, because, as Rizieq himself has said, "in a physical war, winning &

losing is essentially a victory for the *mujahid*, but in a war of thought, winning is an absolute price, because if you lose, faith is at stake" (Sofyan 2018; Arrahmah 2019). In a further attempt to encourage a cult of death and martyrdom, the FPI leader tells his followers that death in the physical world is something trivial and that to die "fighting for Allah is a beauty that is second to none" (Sofyan 2018; Arrahmah 2019). This demand that one should be willing to die for Allah and in the fight against the takeover of Indonesia by infidels encourages FPI supporters to perceive in death something glorious and to seek it out in an attempt to win the admiration of both God and other FPI supporters. FPI supporters thus may not merely be motivated by negative emotions (anger, fear), but by a desire to do good, to win glory, and to win admiration and status in this life and the next.

The Ahok affair provided Rizieq with an opportunity to frame a Christian and Chinese politician as an enemy of 'the people' or the ummah, and present Indonesian elites as unwilling to protect 'the people' from an immoral foreigner. As a Chinese businessman and a Christian, Ahok represented a dual civilizational enemy and his rise to power was framed by Rizieq as, not only a foreign threat to the ummah in Indonesia but also as part of the global threat to Islam presented by non-Islamic civilizational and national entities. When Ahok appeared in a video criticizing the manner in which Qur'an verses were misused by figures such as Rizieq, the FPI leader denounced Ahok as a blasphemer. As a cleric, Rizieq's opinions carried a certain authority. For example, not only did public pressure, some of it initiated by Rizieq, lead to the Indonesian police charging Ahok with blasphemy, but the court trying Ahok relied upon Rizieq's interpretation of the Qur'an and Ahok words when making their decision to sentence the governor to two years in prison (Kompas TV 2017; Allard and Suroyo 2017; Peterson 2020).

Ahok's imprisonment was a great victory for Rizieq. However, the growing power of the FPI led directly to its destruction at the hands of the state. The Indonesian government appears to have been increasingly aware of the threat posed to Pancasila and pluralism in Indonesia by the FPI and similar Islamist movements. In response, the Joko Widodo-led government attempted to arrest Rizieq on pornography charges in 2017, forcing the FPI leader to flee to Saudi Arabia. In a change of fortunes, Ahok was released from prison in 2019 and was shortly after gifted a job running Indonesia's largest energy company. Rizieq was increasingly incensed by these developments and claimed that Ahok was supported by the government "the president [Widodo], the Indonesian police chief, the Armed Forces commander, KPU and KPK, backed by major political parties, and campaigned by the entire national media, together with a number of pollsters funded by the 'nine red dragons'" (Tempo 2019). In other words, Rizieq framed the reversal of fortunes between he and Ahok as part of a broader conspiracy involving Chinese businessmen and government elites that threatens Islam within Indonesia.

Rizieq's ultimate fall came following his return to Indonesia. Perhaps overestimating his own power, Rizieq returned to Jakarta in 2020. The FPI leader did not 'lie low' once back in the country. He began staging large rallies to "defend Islam" and at times attacked Indonesia's president, calling Widodo "a troublemaker and a source of disaster for Muslims" (Cikimm 2018). The COVID-19 pandemic did not prevent Rizieq from rallying his supporters against the government and condemning Ahok. Nor did the FPI respect the rule of law in Indonesia, refusing to submit to Covid testing on the grounds that the law of Allah and the requirement to command good and fight evil was higher than any government mandate to quarantine or submit to COVID testing. When FPI members were found to be spreading COVID, the government finally had the pretext to ban the group entirely, and without upsetting the broader Indonesian public, many of whom were sympathetic to the group's core aim of 'defending Islam', if not their vigilantism. Rizieq was charged with concealing his positive COVID test result, and, in 2021, was sentenced to four years in prison (Kompas TV 2021). That same day, 200 of his supporters were arrested after they attempted to force themselves into the courtroom, in a demonstration of their zeal and devotion to their leader (Jakarta Globe 2021).

## 8. Conclusions

The rise and subsequent collapse of the FPI suggest there has been a partial 'civilizational turn' among Islamist-populist groups in Indonesia, but that mainstream politicians have so far largely prevented civilizational populism from dominating the public sphere. The discourse of FPI leader Rizieq mirrors, in certain respects, the discourses of Turkish President Erdogan and former Pakistani Prime Minister Imran Khan. All three emphasize the importance of unifying the ummah and claim that Western powers are conspiring with either secularist elites or religious minorities within the nation to oppress the people, i.e., Muslims. However, the political success of the FPI occurred without direct involvement with either parliamentary democracy or electoral politics. Unlike Erdogan's Justice and Development Party (AKP) or Khan's Pakistan Tehreek-e-Insaf (PTI), the FPI was largely a street movement that won power by intimidating Indonesian politicians and forcing them to ignore its vigilantism. Equally, the FPI is led by a credentialed Islamic cleric, 'Imam Besar' (grand imam) Habib Rizieq, not a secular politician like Erdogan or Khan. Of course, Rizieq does not claim that he is rebuilding Islamic civilization worldwide by attempting to replace Muslim nation-states with a global caliphate. Rather, he is a nationalist whose civilizational populist rhetoric is confined largely to constructing a narrative in which governing elites are betraying the interests of the authentic Muslim people of Indonesia by allowing the West to spread its immoralities throughout the nation, and by allowing foreign businessmen, particularly Chinese Christians, to dominate the Indonesian economy and keep the authentic Muslim people of Indonesia poor. Rizieq's populist discourse has proven significant insofar as it has succeeded in framing government support for religious freedom, and the economic power of Christian and Chinese businessmen, as part of a world conspiracy against Muslims, which is taking different forms in different places.

Other Islamist groups have also claimed an anti-Muslim conspiracy exists and that the West conspires with local elites to oppress the ummah (Hadiz 2016, 2018; Yilmaz and Morieson 2021, 2022a). However, although the FPI differs from the Indonesian al-Qaeda affiliated groups such as Jemaah Islamiyah and Hizbut Tahrir Indonesia insofar as the group is not interested in establishing a Caliphate state and supports democracy, it has claimed that regions in which Muslims are a majority require Muslim leaders and Islam based systems of governance. According to the FPI, the Indonesian government must encourage Muslims to obey sharia law and forbid non-Muslims to have such high positions in the government. At the same time, it urges Muslims to fear the 'contagion' of Western immoralities, which they claim is spreading throughout the Indonesian archipelago. It has persistently indoctrinated its followers to believe that liberalism, secularism, pluralism, and freedom of religion are spread by infidels in order to destroy the *ummah*.

Rizieq turned remarks by Ahok, a single politician, into a crisis with national and civilizational dimensions by uniting a series of disparate and unrelated problems into a single calamity. While the 'crisis' was immediately caused by a Chinese Christian politician 'blaspheming', the FPI portrayed the Indonesian government as ineffectual and unwilling to protect the ummah from economic exploitation and the corruption of their morals and framed Ahok as a non-Muslim 'enemy' and source of 'evil'. Rizieq's populist discourse was thus aimed at exploiting existing religious divisions within Indonesia, and its success was the product of his ability to elicit deep feelings of religious rage and fear of 'infidels' attacking Islam and corrupting the morals of the ummah, among Indonesian Sunni Muslims. This discourse was aligned with his broader Islamist conception of international politics as a battleground between 'good' and 'evil'. The FPI, in line with its Islamist conception of politics, constructs a populist narrative in which good and evil people may be identified by their religious identities. Sunni Muslims are 'pious' and 'good', yet face an implacable enemy spreading evil and vice throughout the land they should rightfully control as the majority population. This 'enemy' incorporates Christians, non-Sunni Muslims, secularists, and liberal Muslims, all of whom are charged with spreading—or permitting the spread of—decadent Western ideas incompatible with Islam and deleterious to Islamic society throughout Indonesia. Like other populist groups, the FPI claim that they represent 'the

virtuous people' and are fighting for their interests and against corrupt government elites. Yet they add a civilizational dimension to this populist message, insofar as they define this 'virtuous people' as the Muslim ummah and further claim that the ummah everywhere faces attacks from non-Muslim enemies, in particular from the West. The FPI and Rizieq apply this overarching narrative to Indonesian domestic politics and therefore frame the political and social events occurring in Indonesia as part of a wider cosmic battle between the forces of good (ummah) and evil (non-ummah), and between the national and civilizational entities representing each. By telling Indonesian Muslims to identify primarily with their religion, rather than with fellow citizens, the group encourages a degree of transnational solidarity with the wider body of Muslims, and antagonism toward non-Muslim peoples both within and without Indonesia. Thus, while the FPI remains a nationalist movement, its Islamist populism contains within it an inherent civilizationism insofar as they conceive of the world as a battleground between Islam (constructed as a monolithic civilisational entity that is superior to others) and its civilizational enemies. Indonesia's social and economic problems, according to Rizieq and his followers, are the result of Indonesians' refusal to obey the teachings of Islam, and of the state's neglecting to command good and forbid evil. Islam is thus presented as a solution to Indonesia's problems. The establishment of an Islamic society in Indonesia is therefore portrayed as a noble goal for which FPI followers must fight. FPI supporters are taught to feel hopeful that Islam will solve their nation's problems. However, if they should die 'defending' Islam from the forces of 'evil', and while protecting the ummah from the spread of Western decadence, they will receive a reward in the next life.

The FPI and its 'Imam Besar' have constructed a populist narrative incorporating Islamism and its inherent transnational and civilizational elements, and in which Indonesian politics is framed as a part of a wider battle between good and evil, or between the ummah and its religio-civilizational enemies. This narrative is used to create demand for Islamist populism within Indonesia via an emotional narrative that claims that the 'ummah' is threatened by evil non-Muslim forces and must fight to defend Islam, and ultimately to justify and legitimize defiance of legitimate state authority, and violence against vulnerable non-Sunni minority groups. The rise and fall of the FPI is a demonstration of the growing significance of Islamism and populism in Indonesia, and of the battle occurring within Indonesia between those who believe Islam and Islamic civilization are inherently plural, and groups such as the FPI who despise and attack pluralism. The banning of the FPI, jailing of Rizieq, and rehabilitation of Ahok after his imprisonment may give the impression that the pluralists are winning this battle. Yet this may not be the case. The growing conservatism of the MUI, the barely concealed support Rizieq and other Islamist groups enjoy within the state and its apparatuses, and indeed the necessity of banning the FPI due to its increasing power over political and religious discourse suggests that Indonesia may be becoming increasingly intolerant of non-Sunni expressions of religiosity in the public sphere. Equally, the media coverage of the Ahok affair, which suggested at times that it represented a clash of civilizations between Islam, the West, and China, further suggests that there is an appetite in Indonesia for narratives asserting multiple civilizational threats to the ummah (Ritonga et al. 2020).

**Author Contributions:** Conceptualization, I.Y. and N.M.; methodology, I.Y. and N.M.; investigation, I.Y., N.M. and H.B.; writing—original draft preparation, I.Y. and N.M; writing—review and editing, I.Y., N.M. and H.B; project administration, I.Y.; funding acquisition, I.Y. All authors have read and agreed to the published version of the manuscript.

**Funding:** Australian Research Council: DP220100829 Religious Populism, Emotions and Political Mobilisation.

**Data Availability Statement:** Not applicable.

**Conflicts of Interest:** The authors declare no conflict of interest.

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
