# Peer review of "Civilizational Populism in Indonesia: The Case of Front Pembela Islam (FPI)"

_religions, doi:10.3390/rel13121208_

Round 1

Reviewer 1 Report

This is a scholarly and timely paper which will be of interest to readers of this journal who have an interest in populism and religion. However, there is a lot of unnecessary repetition in the paper, which should be removed. I also found the points of argument circular rather than logically sequential. The linear structure of the paper could be much improved without losing the valuable content. There are many mistakes in the writing which need to be corrected. Overall, a solid contribution to the field.

Author Response

Thank you very much for your strong support. We have now deleted the repetitive and circular content of the paper and have corrected the typos.

  1. We have removed repetitious material describing the FPI’s history and the Ahok case.
  2. We have made alterations to the structure, including removing repetitious material and given each section a thematic focus previously lacking.
  3. We have edited the paper to remove the spelling and grammatical mistakes.

Reviewer 2 Report

The general concept of the article is good and the content is interesting. Nevertheless,  there is a lack of a deep analysis of why ultra-right groups gain popularity in society. Consequently, the text gives the impression that the issue has been presented one-sidedly, without taking into account nuances, such as the fact that far-right ideas fit into a certain social vacuum, a niche that is filled by populistic ideas pretending to tackle the social and economic problems such as social inequalities, exclusion, poverty, etc. By showing those aspects, the author will give a more comprehensive picture of Indonesia's political scene and Islam.

Author Response

Thank you very much for your strong support of the paper. We have tried to address this issue. We have explained that while there are many groups involved we are only focusing on one of these groups.

Reviewer 3 Report

In general, the article is very well structured. The main idea of ​​the article is the study of the populist drift of politics in Indonesia. The author states at the beginning of the article that in Indonesia the idea of ​​civilizing populism dominates in political discourse, that is, the use of a discourse of fragmentation of Indonesian society between good and bad, the former being Sunni Muslims and the latter all those considered heretics or who identify with the values ​​of Western civilization.

To begin with, the author makes a good approach to the literature that he will use. He names well the latest works by Yilmaz, who has published several articles on the study of the relationship between populism and Islam, and also talks about the main works on the populist movements of recent years (Brubaker, Mietzner).

Once the theoretical framework has been exposed, the author makes a good exposition of the precedents for the domination of populist discourse in Indonesian politics, from the political opening after the end of the dictatorship to the less pluralistic trend of today. To explain it, the author takes as referents the role of the FPI Islamic movement in Indonesian politics and the political importance of the Ahok case. The article is itself a journey into the advance of the FPI's influence and the importance it played in the outcome of the Ahok case, which completely transformed Indonesian political life. The author's conclusion is that, after the case, a civilizing populist discourse prevails in Indonesia that the political elites have instrumentalized for their benefit.

The article, in short, is well structured and argued. I don't see noteworthy issues that the author should add, except for certain formal defects in some bibliographical references that he should review (lines 50, 336).

Author Response

Thank you very much for your strong support of the paper. Thank you for your positive evaluation. In line 336, we changed it to (Barton 2021; Fossati and Mietzner 2019; Aspinall and Mietzner 2019; Rosadi 2008; Syihab 2008).

Reviewer 4 Report

Dear authors, 
I am glad to read and review your manuscript. It indicates the authors' mastery of the populism study. 

Nevertheless, it requires a lot of improvement in several aspects. Following is my comment: 
1. The authors capture a broad topic, therefore, the research focus is vague. The research seems to try to capture a variety of civilizational populism of FPI, however, at the same time, it's interested in Ahok case. Focus on Ahok case should be considered so that so don't need to capture everything about FPI. 
2. Research scope - The study states post-Suharto, unfortunately, it doesn't adequately variety of events, changes, and dynamics of FPI from 1998 till today. Avoid "post-Suharto" if you want to take a specific event or topic. 
3. Organization - it is a little bit difficult to follow findings and discussion. It is recommended to present a thematic format, e.g. section 1: FPI and populism; section 2: FPI and political discourse, etc. 
4. It doesn't describe FPI's connection to ISIS, which was shown by Rizieq in a video and became a justification for the government to dismiss the group. The involvement of its former spokesperson in ISIS' activities should also be described. 
5. Inconsistency in citation formatting must be avoided. 
6. Some question remarks indicate the authors were not sure about what they wrote. It must be avoided. Working with a professional proofreader would help the writing process. 
7. Research methodology is not available. This aspect must be available since the journal and most journals appraise this aspect. 

Detailed note is available in the attached document.  

Best regards

Author Response

For the detailed responses to the reviewer's comments, please see the attachment. Many thanks for your comments. 
